# A Novel Multi-Sensor Nonlinear Tightly-Coupled Framework for Composite Robot Localization and Mapping

**DOI:** 10.3390/s24227381

**Published:** 2024-11-19

**Authors:** Lu Chen, Amir Hussain, Yu Liu, Jie Tan, Yang Li, Yuhao Yang, Haoyuan Ma, Shenbing Fu, Gun Li

**Affiliations:** 1School of Aeronautics and Astronautics, University of Electronic Science and Technology of China, Chengdu 611731, China; lchen@std.uestc.edu.cn (L.C.); liuyuuestc2022@163.com (Y.L.); tanjie@std.uestc.edu.cn (J.T.); yli@std.uestc.edu.cn (Y.L.); yangyh@std.uestc.edu.cn (Y.Y.); 202222100306@std.uestc.edu.cn (H.M.); 202221100304@std.uestc.edu.cn (S.F.); 2School of Computing, Edinburgh Napier University, Scotland EH10 5DT, UK

**Keywords:** composite robots, multi-sensor fusion, nonlinear tight coupling, SLAM, illuminance conversion

## Abstract

Composite robots often encounter difficulties due to changes in illumination, external disturbances, reflective surface effects, and cumulative errors. These challenges significantly hinder their capabilities in environmental perception and the accuracy and reliability of pose estimation. We propose a nonlinear optimization approach to overcome these issues to develop an integrated localization and navigation framework, IIVL-LM (IMU, Infrared, Vision, and LiDAR Fusion for Localization and Mapping). This framework achieves tightly coupled integration at the data level using inputs from an IMU (Inertial Measurement Unit), an infrared camera, an RGB (Red, Green and Blue) camera, and LiDAR. We propose a real-time luminance calculation model and verify its conversion accuracy. Additionally, we designed a fast approximation method for the nonlinear weighted fusion of features from infrared and RGB frames based on luminance values. Finally, we optimize the VIO (Visual-Inertial Odometry) module in the R3LIVE++ (Robust, Real-time, Radiance Reconstruction with LiDAR-Inertial-Visual state Estimation) framework based on the infrared camera’s capability to acquire depth information. In a controlled study, using a simulated indoor rescue scenario dataset, the IIVL-LM system demonstrated significant performance enhancements in challenging luminance conditions, particularly in low-light environments. Specifically, the average RMSE ATE (Root Mean Square Error of absolute trajectory Error) improved by 23% to 39%, with reductions from 0.006 to 0.013. At the same time, we conducted comparative experiments using the publicly available TUM-VI (Technical University of Munich Visual-Inertial Dataset) without the infrared image input. It was found that no leading results were achieved, which verifies the importance of infrared image fusion. By maintaining the active engagement of at least three sensors at all times, the IIVL-LM system significantly boosts its robustness in both unknown and expansive environments while ensuring high precision. This enhancement is particularly critical for applications in complex environments, such as indoor rescue operations.

## 1. Introduction

With the rapid advancement of robotic technology, autonomous mobile robots, vehicles, and drones are increasingly being applied across various sectors, including industry, military, disaster response, space exploration, and domestic services [1,2,3,4,5]. For robots to achieve autonomous navigation and intelligent interaction, precise mapping and localization technologies—known as SLAM (Simultaneous Localization and Mapping)—are especially crucial. This issue primarily encompasses two domains: localization and mapping. Localization refers to accurately determining a robot’s position and orientation within its environment, while mapping involves integrating partial observations of the surrounding environment. Initially, researchers treated localization and mapping as separate challenges; however, they later discovered that these processes are tightly coupled. An accurate map is necessary for a composite robot to orient itself in its current environment, and precise positioning is essential to construct a reliable map. However, the presence of various types of measurement noise and disturbances in real-world environments significantly complicates the resolution of SLAM problems. This complexity makes the fusion of multiple sensors critical. By integrating multiple sensors, uncertainties within SLAM can be effectively addressed [6,7,8].

Building on the foundation of R3LIVE++ [9], this paper addresses the problem of precise state estimation and mapping for composite robots in semi-enclosed, unstructured areas. It focuses on overcoming the challenges posed by traditional single sensors related to lighting and interference, significantly enhancing the accuracy and reliability of environmental perception and self-positioning measurements. The approach involves nonlinear interpolation and weighted fusion of multiple sensor data and feature types, facilitating three-dimensional spatial perception, modeling, calibration, matching, and measurement. A key emphasis is placed on addressing the challenges posed by significant optical changes, particularly in low-light or dark environments. This focus ultimately enables the autonomous navigation of mobile robots in unknown settings. The main contributions of our work can be summarized as follows:We propose a luminance conversion model based on RGB images. The model utilizes mean values from multiple samples of RGB images taken under maximum and minimum illumination conditions in a simulated indoor rescue scenario. These mean values are then converted to serve as normalized standard values.We perform nonlinear interpolation and weighted fusion of visual sensor data based on real-time luminance values. The images undergo processing where two types of features are weighted and fused. The allocation of weights dynamically changes according to thresholds set for luminance values, ensuring tight coupling of multi-sensor data.We introduce IIVL-LM, a tightly coupled composite localization and mapping system for composite robots, utilizing the R3LIVE++ framework. This system integrates an IMU, visual sensors (infrared and RGB), and LiDAR, as illustrated in Figure 1. Additionally, by integrating the depth information acquisition capabilities of infrared cameras, we have optimized the VIO fusion module.We conducted extensive experiments under simulated indoor rescue scenarios and with the TUM-VI dataset [10], comparing our framework with similar systems.

## 2. Related Work

### 2.1. LiDAR-Inertial SLAM

Recent advancements in LiDAR technology, characterized by significant improvements in performance and reliability alongside reduced costs, have fueled extensive research into LiDAR SLAM. Zhang et al. presented a real-time LiDAR odometry and mapping system called LOAM (LiDAR Odometry and Mapping in Real-time) [11]. This innovative framework employs scan-to-scan registration for precise localization and scan-to-map techniques for comprehensive mapping. Additionally, it integrates an IMU in a loosely-coupled manner to rectify LiDAR scan distortions without involving the IMU bias in the scan registration process. Shan et al. further refined this approach by introducing LeGO-LOAM (Lightweight and Ground-Optimized LiDAR Odometry and Mapping on Variable Terrain) [12], a lightweight variant that discards unreliable features during the ground plane segmentation phase. This optimization enables the algorithm to operate in real-time on platforms with limited computational resources. Notably, these advancements primarily cater to multi-line spinning LiDARs. In contrast, solid-state LiDARs are characterized by irregular scanning patterns and narrower fields of view. Prior research has predominantly relied on scan-to-map registration for both localization and mapping tasks in these LiDARs [13]. When compared with loosely-coupled methods, tightly-coupled approaches exhibit superior robustness and accuracy. A prime example is LIO-SAM (LiDAR Inertial Odometry VIA Smoothing and Mapping) [14], which enhances precision by optimizing a sequence of keyframe poses within a factor graph. Another groundbreaking development is LINS (A LiDAR-Inertial State Estimator for Robust and Efficient Navigation) [15], the first system to address the 6-DOF (Degrees of Freedom) ego-motion challenge through iterated Kalman filtering. This approach is implemented in a tightly-coupled LIO (LiDAR-Inertial Odometry) system. FAST-LIO [16] reduced the computational demands of calculating the Kalman gain by introducing a formula where the computational complexity is tied to the state dimension rather than the measurement dimension. This innovation significantly lowers the computational burden. Its successor, FAST-LIO2 [17], further improved efficiency by introducing an incremental k-d tree. This advancement utilizes raw point configurations to capture finer environmental details, ultimately bolstering localization accuracy and robustness.

### 2.2. Visual-Inertial SLAM

According to the classification in [18], visual SLAM methodologies are broadly categorized into two types: indirect and direct methods. Indirect methods, also referred to as feature-based methods, involve feature extraction, data association, and the minimization of feature reprojection errors. Conversely, direct methods minimize photometric errors—or intensity differences—between consecutive images. Within the realm of indirect methods, notable systems include Davison et al.’s MonoSLAM (Monocular Simultaneous Localization and Mapping) [19]. This system recovers the camera’s 3D trajectory in real-time by creating a sparse yet persistent natural landmark map within a probabilistic framework. Klein and Murray’s PTAM (Parallel Tracking and Mapping) [20] innovates by separating tracking and mapping into parallel threads and efficiently selects visual landmarks from a subset of frames for bundle adjustment (BA) optimization. This approach allows for the estimation of camera poses and landmark positions. ORB-SLAM (Oriented Fast and Rotated Brief Simultaneous Localization and Mapping) [21], a more comprehensive and reliable framework, utilizes consistent functions across tasks such as tracking, mapping, relocalization, and loop closure. Its successor, ORB-SLAM2 [22], expands its capabilities to support monocular, stereo, and RGB-D (Red, Green, Blue, and Depth) cameras. Both VINS-MONO (Visual-Inertial System Monocular) [23] and ORB-SLAM3 [24] address the issue of scale ambiguity in visual SLAM by integrating IMU measurements and image features within sliding window bundle adjustment optimizations, ensuring high-precision localization.

On the other hand, direct visual SLAM methods, also known as photometric-based methods, excel in low-texture environments, demonstrating superior short-term performance. These methods have found success in 2D sparse feature tracking, similar to the Lucas–Kanade optical flow [25], and have been extended to visual SLAM. Engel et al.’s LSD-SLAM (Large-Scale Direct Monocular Simultaneous Localization and Mapping) [26] represents a direct monocular SLAM algorithm that relies on image intensities for tracking and mapping. It employs direct image alignment for incremental camera pose tracking and performs pose graph optimization to preserve global consistency. In DSO (Direct Sparse Odometry) [18], the authors introduced a direct probabilistic model fully integrated with comprehensive photometric calibration, combining photometric bundle adjustment for optimal accuracy and robustness. To achieve real-time performance when using a standard CPU (Central Processing Unit), they also capitalized on the sparse structure of the corresponding Hessian matrix. The photometric model offers precise pose estimation for short-term tracking, eliminating the need for data association. Mixed methods, which utilize both photometric and geometric errors, have also emerged. A notable example is SVO (Semi-Direct Visual Odometry) [27] by Forster et al. In SVO, minimizing photometric errors resolves short-term tracking issues, while long-term drift is mitigated through windowed bundle adjustment on visual landmarks.

### 2.3. LiDAR-Visual Fusion SLAM

Building upon LiDAR-inertial methods, the integration of visual sensor measurements in LiDAR-inertial-visual odometry systems offers superior robustness and precision. Zhang and Singh [28] proposed a system that incorporates a loosely-coupled VIO as a motion model to initialize the LiDAR mapping subsystem. In another study [29], a fusion approach was introduced that combines tightly-coupled stereo VIO with LiDAR mapping and LiDAR-enhanced visual loop closure. Despite this integration, the overall fusion remains loosely-coupled because LiDAR data were not jointly optimized with visual-inertial data. For heightened precision and robustness, recent frameworks have proposed tightly-coupled sensor data fusion. Zuo et al. [30] presented an LIC (LiDAR-Inertial-Camera) fusion framework, which integrates IMU readings, sparse visual features, and LiDAR plane and edge features. This system, based on the MSCKF (Multi-State Constraint Kalman Filter) framework, demonstrated experimental results that surpassed other modern methods in terms of accuracy and robustness. An Y et al. proposed a Visual-LiDAR SLAM method based on unsupervised multi-channel deep neural networks in 2022 [31], addressing the challenge of accurate mapping and localization in complex environments without the need for labeled data. Shan et al. [32] introduced LVI-SAM, a tightly-coupled integration of LiDAR, visual, and inertial sensors, founded on a factor graph. Its subsystems can operate independently or jointly, depending on the availability of features. R2LIVE, on the other hand, integrates LiDAR and camera data within a Kalman filter, showing resilience in various challenging scenarios.

These LiDAR-inertial-visual systems predominantly concentrate on localization, paying limited attention to map efficiency and precision. Consequently, their visual and LiDAR subsystems often maintain separate LIO and VIO maps, impeding deeper data fusion and high-precision colored 3D map reconstruction. R3LIVE++ strives to achieve real-time localization and radiometric map reconstruction, with a shared radiometric map at its core, maintained by both LIO and VIO subsystems. Specifically, the LIO subsystem constructs the map’s geometric framework, while the VIO subsystem restores its radiometric details. In R3LIVE++, both LIO and VIO subsystems employ direct methods. Bundle adjustment reduces long-term variations in both the LIO subsystem, which is based primarily on FAST-LIO2 [17], and the VIO subsystem, which depends on photometric measurements. This alignment technique, which aligns frames to the map, effectively diminishes range drift while maintaining computational efficiency.

## 3. Nonlinear Tightly-Coupled Model

### 3.1. Overview of IIVL-LM System Framework

This paper refines the VIO within the R3LIVE++ fusion framework, specifically addressing the issue of feature loss in RGB cameras due to complete darkness. The VIO module is divided into two sub-modules: one for the infrared camera and one for the RGB camera. Through the Illuminance Conversion Model module, luminance is calculated in real-time to achieve effective, tight coupling between the infrared and RGB frames. The NIKFF (Non-linear Insert Keyframe Feature Fusion) module effectively segments the luminance into three non-linear stages. In the middle stage, weighted fusion is used to extract image features. Additionally, the Ceres Solver [33] method is employed to solve the non-linear optimization interpolation problems within this module. The system configuration is depicted in Figure 2.

### 3.2. Basic Sensors Fusion Theory

Multi-sensor fusion is primarily categorized into two subsystems: LIO (LiDAR and IMU) fusion and VIO (Vision and IMU) fusion. In the LIO subsystem, the geometric structure of the global map is reconstructed using data from LiDAR scans. The IMU is employed to provide motion compensation for the *k*-th observation point detected by the LiDAR:(1)P Gs=Rˇ GIkR ILP Ls+P IL+Pˇ GIk

P Gs represents the position of point *P* in the global coordinate system. R IL and P IL are the rotation matrix and offset, respectively, which are used to transform coordinates from the LiDAR coordinate system to the IMU coordinate system. The first step involves converting the raw point coordinates, as directly observed by the LiDAR, into coordinates within the IMU coordinate system. This conversion achieves motion compensation for the original points. Rˇ GIk and Pˇ GIk denote the rotation matrix and offset for transforming from the IMU coordinate system to the global coordinate system, respectively. The second step then completes the transformation from the IMU coordinate system to the global coordinate system.

Next, achieving precise point cloud matching to the global map involves searching for the five nearest neighbor points to fit a plane. To improve the efficiency of searching for the nearest neighbor points, we process the map points using a k-d tree [17]. Although three points are sufficient to determine a plane, in practical implementation, we choose to use five points to improve the robustness of plane fitting. Due to noise in LiDAR point clouds and the complexity of the environment, relying on only three points may result in a less accurate fit. Using five points for plane fitting can reduce the impact of noise and enhance the accuracy of the fit. When these points are not perfectly coplanar, we calculate the residuals of each point to the fitted plane and use the LMSE (Least Mean Square Error) method to ensure the optimal plane is selected [34]. Finally, the normal vector $u_{s}$ and centroid $Q_{s}$ of this plane are determined, then the LiDAR measurement residuals are then calculated:(2)rlXˇkPsL=usTP Gs−Qs

In this context, $Q_{s}$ represents the centroid and $u_{s}$ is the normal vector of the plane. The measurement error is assessed by calculating the distance from the measurement points to the fitted plane. Ideally, the residual should be zero. However, due to estimation errors and LiDAR measurement noise, the residuals are usually non-zero. We employ a manifold error state iterative Kalman filter framework, which updates the estimated state by solving the optimization problem of state errors. As a result, this process brings the estimated state closer to the true state. The true state $x_{k}$ is combined with the prior distribution from the IMU propagation, as shown in Equation (3):(3)minδxˇk⁡(∥(xˇk⊞δxˇk)⊟x^k∥∑δx^k 2+∑s=1m∥rlxˇk,P Ls+Hslδxˇk∥∑αs 2
where ${\left\|x\right\|}_{\sum }^{2}=x^{T}\sum^{-1}x$ is the squared Mahalanobis distance with covariance ∑, ${\hat{x}}_{k}$ is squared Mahalanobis distance with covariance Σ, ${\hat{x}}_{k}$ is the IMU propagated state estimate, and $\sum_{\delta {\hat{x}}_{k}}\;$ is the IMU propagated state covariance. ⊞ and ⊟ symbols represent the effects on different error terms in the state update, specifically the propagation of error between the state variable and the observation. For further details, please refer to Section IV-E of R2LIVE [35].

During the reconstruction of the global map’s geometric structure by the VIO subsystem, radiometric information is extracted from the input RGB images. Assuming there are M map points $P=\{{Ρ}_{1},\ldots ,{Ρ}_{m}\}$, their projections in the frame $I_{k-1}$ are $\rho =\{\rho_{1_{k-1}},\ldots ,\rho_{m_{k-1}}\}$. The Lucas–Kanade optical flow technique is applied to locate these points in $I_{k}$, and the estimated state ${\overset{ˇ}{x}}_{k}$ is computed during the current iteration to determine the reprojection error:(4)rcxˇkρsk,P GS=ρsk−πP Gs,xˇk

πP Gs,xˇk represents the predicted position of the pixel, which can be calculated using the following Equation (5):(5)πP Gs,xˇk=πphP Gs,xˇk+tˇCkI∆tk−1,kρsk−ρsk−1

Here, tˇCkI∆tk−1,k represents the time correction parameter. tˇCkI is the time offset between the IMU and the camera. πphP Gs,xˇk is the predicted pixel position calculated using the standard pinhole camera model. $∆t_{k-1,k}$ represents the time interval between the previous image and the current image. Similar to updates in the LIO system, the measurement noise in the VIO system originates from two sources: pixel tracking error and map point localization error. These types of noise can be corrected using Equation (6) to approach the true state:(6)0=rcxk,ρskgt,PsgtG≈rcxˇk,ρsk,PSG+Hslδxˇk+βs

The formula above establishes the observation distribution, which is combined with IMU propagation to derive the MAP (Maximum A Posteriori) estimate of the state. This method is analogous to the LIO update approach. The converged state estimates are then utilized to refine the VIO from frame-to-map alignment, ensuring precise and consistent mapping and localization.

Frame-to-frame VIO updates provide robust state estimates, which are further refined through frame-to-map VIO updates by minimizing the radiometric error of the tracked map points *P*. Using the current iteration’s state estimates, the tracked map points are projected onto the image plane. These state estimates include the estimated camera pose, intrinsic and extrinsic parameters, and exposure time. This projection determines the pixel positions of the tracked map points. The radiometric error can be calculated using Equation (7):(7)rcxˇk,PSG,Υs=Φs−Υs,Φs=ϵˇkΓkρˇsk

First-order Taylor expansion of real residuals:(8)0=rcXk,P Gsgt,γsgt≈rcxˇk,P Gs,cs+Hscδxˇk+ζs

Similar to earlier, Equation (8) forms the observational distribution for the state, which, when combined with IMU propagation, yields the MAP estimate of that state.

After the frame-to-map VIO update, the precise pose of the current image is obtained. A Bayesian update is then performed to determine the optimal radiance for all map points, minimizing the average radiance error between each point and its corresponding image.

Next, all points within the activated voxels are retrieved. Assume the set of retrieved points is *Q*. For the *s*-th point within the current image’s FoV (Field of View), the observed radiance vector is first obtained. This is achieved by calculating the frame-to-map radiance error, after which the covariance is computed. If it is a new point added by the LIO subsystem, we set:(9)$$\Upsilon_{s}=\Phi_{s},\;\sum_{n_{\gamma_{s}}}\;=\sum_{n_{\Phi_{s}}}\;$$

Finally, the radiation vector $\Upsilon_{s}$ stored in the map is fused with the newly observed radiation vector $\Phi_{s}$ and covariance $\sum_{n_{\Phi_{s}}}\;$ by Bayesian updating. Equations (10) and (11) are used to calculate the new covariance matrix $\sum_{n_{{\tilde{\Upsilon }}_{s}}}\;$ and the updated radiation vector ${\tilde{\Upsilon }}_{s}$. Equation (12) is then used to reassign the updated radiation vector and covariance matrix back to their respective locations. By repeatedly fusing the observed radiation data, radiation errors are reduced, thus obtaining the optimal radiation values.
(10)$$\sum_{n_{{\overset{ˇ}{\gamma }}_{s}}}={\left({\left(\sum_{n_{\gamma_{s}}}\;+\sigma_{ic}^{2}\cdot ∆t_{\gamma s}\right)}^{-1}+\sum_{n_{\Phi_{s}}}^{-1}\;\right)}^{-1}$$
(11)$${\tilde{\Upsilon }}_{s}={\left({\left(\sum_{n_{\gamma_{s}}}\;+\sigma_{ic}^{2}\cdot ∆t_{\gamma s}\right)}^{-1}\gamma s+\sum_{n_{\Phi_{s}}}^{-1}\;\Phi_{s}\right)}^{-1}\sum_{n_{{\tilde{\Upsilon }}_{s}}}\;$$
(12)$$\Upsilon_{s=}{\tilde{\Upsilon }}_{s},\sum_{n_{\gamma_{s}}}\;=\sum_{n_{{\tilde{\Upsilon }}_{s}}}\;$$

### 3.3. Illuminance Conversion Model

Given the real-time requirements and the fact that luminance does not change abruptly over a short period, luminance detection can be performed using fixed-interval frame sampling based on the application scenario. Alternatively, it can be achieved by synchronously evaluating key frames extracted during loop closure detection. The conversion is shown in Equation (13):(13)$$E=\frac{k_{r}R+k_{g}G+k_{b}B}{Y_{\max}}$$

In this context, *R*, *G*, and *B* represent the values of the RGB channels of an image, and $k_{r}$, $k_{g}$, and $k_{b}$ represent the coefficients of the *R*, *G*, and *B* channels, respectively. The channel coefficients are set at 0.299, 0.587, and 0.114, respectively. *Y_max_* is the maximum value for the RGB channels, which is 255 for an 8-bit RGB image.

The infrared camera in this study is capable of outputting two types of images, enabling automatic timestamp frame alignment. The luminance value of the RGB images is checked every 60 frames, matching the frame rate of both cameras. Take the *i*-th frame as the test frame and use the values from the two frames before and after the *i*-th frame for a comprehensive evaluation. After removing the maximum and minimum values, calculate the average of the remaining values; the luminance $E_{i}$ of the test frame is calculated as follows:(14)$$E_{i}=\frac{\sum_{j\in \left\{i-2,i-1,i,i+1,i+2\right\}}E_{j}-\max(E_{j})-\min(E_{j})}{3}$$

Using samples collected under the maximum and minimum indoor luminance for normalization, with an average maximum luminance of 50,287 lux and a minimum of 21 lux, a total of 48 frames yields $E_{\max}$ and $E_{\min}$. Thus, the normalized luminance value of the detection frame is calculated as follows:(15)$${\hat{E}}_{i}=\frac{E_{i}-E_{\min}}{E_{\max}-E_{\min}}$$

### 3.4. Nonlinear Feature Weight Allocation Method

The performance of infrared and RGB images varies significantly under different lighting conditions. Studies have shown that infrared images are susceptible to overexposure, noise, and poor signals in high-brightness environments, which degrades image quality. Conversely, in very low-brightness conditions, the signal-to-noise ratio of RGB images drops sharply, a limitation also commonly noted in other visual localization studies [36,37]. By selecting appropriate normalized illuminance thresholds, we ensure a smooth transition of weights between the two sensors under dynamically changing lighting conditions, thus avoiding system instability due to the failure of a single channel [38]. This thresholding approach has also been applied in other multimodal visual sensing studies, validating its scientific effectiveness [39,40]. Based on a nonlinear distribution determined by value, we dynamically allocate the infrared image weight α and the RGB image weight β, as shown in Equation (16):(16)$$(\alpha ,\;\beta )=\left\{\begin{array}{l} {\hat{E}}_{i}\geq 0.85;\alpha =0,\;\beta =1\\ 0.15\leq {\hat{E}}_{i}\leq 0.85;\;\alpha =1-{\hat{E}}_{i},\;\beta ={\hat{E}}_{i}\\ {\hat{E}}_{i}\leq 0.15;\alpha =1,\;\beta =0 \end{array}\right.$$

We determined the thresholds through experiments by testing the robustness of the system under various lighting conditions. In extreme illuminance scenarios, we performed feature extraction on images captured by both cameras and analyzed their effectiveness. It was observed that when the normalized luminance conversion value exceeds 0.85, the utility of infrared imagery in the visual localization system significantly diminishes, leading to its discontinuation within the system. Conversely, when this value falls below 0.15, RGB imagery approaches a threshold of failure and is consequently discarded. The experimental results are shown in Figure 3. As illustrated in the figure, when the normalized illuminance value is 0.148, the RGB image contains almost no valuable feature points. Similarly, at an illuminance value of 0.853, the infrared image encounters the same issue. For values between these two thresholds, the system employs a linear strategy to allocate feature values. This method aligns with the LMSE principle. By assigning appropriate weights within the threshold range, it reduces fluctuations in image quality under extreme lighting conditions, thereby ensuring the overall stability of the system [41].

Due to the requirement to extract features from both infrared and RGB images simultaneously, we opt to use a Box filter. This filter approximates the calculation of the Hessian [42] values at each point in the image as image features, aiming to accelerate the process. Consequently, when constructing the image pyramid at different Gaussian template scales σ, the feature value at point x=(x,y) is calculated as follows:(17)$$H(x,\sigma )=\left[\begin{array}{l} L_{xx}(x,\sigma )\\ L_{yx}(x,\sigma ) \end{array}\right.\left.\begin{array}{l} L_{xy}(x,\sigma )\\ L_{yy}(x,\sigma ) \end{array}\right]$$

In this context, $L_{xx}(x,\sigma )$ represents the convolution of the image at location x with a second-order Gaussian template $\frac{{𝝏}^{2}}{𝝏x^{2}}g(\sigma )$, and similar operations apply to other calculations. By using a Boxfilter for convolution approximation, elements in the integral image can be quickly accessed to expedite the computation process. Thus, the approximate Hessian value for each pixel is:(18)$$\det(H)=D_{xx}*D_{yy}-0.85*D_{xy}^{2}$$

In this description, D represents the approximate value of the second-order Gaussian derivatives in different directions after convolution with a Boxfilter. To balance the errors introduced by using the Boxfilter, a weighting factor of 0.85 is applied in the $D_{xy}$ direction. Consequently, the formula for fusing the weighted infrared feature $\det{\left(H\right)}_{I}$ and RGB feature $\det{\left(H\right)}_{R}$ is as follows:(19)$$H_{w}=\alpha *\det{\left(H\right)}_{I}+\beta *\det{\left(H\right)}_{R}$$

Thus, the detailed VIO module, when reconstructing the global map, simultaneously recovers radiometric information from the input RGB and infrared images. Then, *M* map points, represented as ${𝒫}_{H_{w}}\{H_{w}{Ρ}_{1},\ldots ,H_{w}{Ρ}_{m}\}$ after weighted feature fusion, are projected as $\rho_{H_{w}}=\{{H_{w}\rho }_{1_{k-1}},\ldots ,H_{w}\rho_{m_{k-1}}\}$ in frame $I_{k-1}$. By substituting these into Equations (20) and (21), the projection error and radiometric error, respectively, are calculated as follows:(20)rcxˇk,ρHwsk,P GS=ρHwsk−πP Gs,xˇk
(21)rcxˇk,PSG,Υs=Φs−Υs,Φs=ϵˇkΓkρˇHwsk

The above completes the visual refinement fusion work within the VIO module. Additionally, during loop closure detection, this paper relies on dynamic weights to optimize the interpolation values of the three sensors. Since illumination does not affect the performance of LiDAR, the interpolation ratio for LiDAR remains unchanged. However, the interpolation for RGB and infrared images changes linearly according to the variation in weights, as shown in Figure 4. Different colors represent frames from different sensors, with solid lines representing key frames for loop closure detection and dashed lines representing ordinary frame information. This variation indicates that the number of frame insertions for sensor type varies depending on the weights applied during global mapping under different illumination conditions.

### 3.5. VIO (Visual and IMU Fusion) Odometry Based on Stereoscopic Information

After effectively integrating the infrared and RGB cameras and completing the visual frontend work, we optimized the VIO module in the R3LIVE++ framework based on the camera’s capability to acquire depth information. We established a system model that considers the unique mobility characteristics of composite robots, integrating the respective features of the IMU and VO. The optimized visual-inertial odometry model is depicted in Figure 5.

In the process of optimizing VO to handle new IMU data, the current system state is first assessed through filter state prediction. This prediction is then augmented with the new image frame after timestamp alignment to determine if new keyframe tracking is necessary. When the system identifies *NNKF* = 1 (Need for a New Key Frame), feature point tracking is executed to identify and track known feature points across successive image frames. This process includes a two-layer filtering approach. First, it checks for feature point loss. If feature points are lost, indicating the need to track additional points to meet system requirements, new feature points are extracted. These newly extracted feature points are homogenized to ensure uniform distribution in the image space, providing more comprehensive and accurate information. Next, as additional feature points are introduced as observations, the system calculates the observed count as *N* + 1 and enters the depth convergence phase. We use a deep filter to process and update each feature point’s depth information, leveraging multi-frame data for more accurate depth estimation.

The depth information for feature points is calculated using a fifth-order convergent Newton–Raphson method [43,44], iteratively approximating the optimal estimate of the state variables. The input consists of a depth equation system F, constructed using projection equations and geometric constraint conditions, along with the ground robot’s pose information. The output includes the depth values of the feature points and the final spatial coordinates of the target. The specific iterative equation is as follows:(22)$$Y^{n}=X^{n}-\frac{F\left(X^{n}\right)}{F^{\prime }\left(X^{n}\right)}$$
(23)$$Z^{n}=X^{n}-\frac{2F\left(X^{n}\right)}{F^{\prime }\left(X^{n}\right)+F^{\prime }\left(Y^{n}\right)}$$
(24)$$X^{n+1}=Z^{n}-\frac{F\left(Z^{n}\right)}{F^{\prime }\left(Y^{n}\right)}$$

Here, $X^{n}$ is the vector to be solved, composed of the 3D depth estimation constraint equation $P_{wzi}$, which is derived from the depth camera’s intrinsic parameters, projection model, and the pixel coordinates of the feature points. $Y^{n}$ and $Z^{n}$ are intermediate variables used in the iteration process. Considering practical engineering applications, the initial values of all unknowns in the $X^{n}$ vector are set to 200, assuming that the target feature points are located approximately 2 m in front of the composite robot. In actual testing, it was found that setting the initial values between 0 and 800 has little impact on the algorithm’s accuracy and speed. The termination condition for the iterative algorithm is as follows:(25)$${\left\|X^{n+1}-X^{n}\right\|}_{1}<0.01\;\mathrm{or}\;\eta \geq 20$$

Here, η represents the maximum number of iterations.
(26)$$\left\{\begin{array}{l} P_{wxi}=P_{cxi}\cdot P_{wzi}\\ P_{wyi}=P_{cyi}\cdot P_{wzi} \end{array}\right.\;\;\;i=1,2,3,4$$

By substituting the solution of $P_{wzi}$ into Equation (26), the spatial coordinates of the target feature points in the camera $P_{w}i$ coordinate system can be obtained. Then, by substituting the pose R and T into the coordinate transformation equation, the target feature points’ world coordinates can be computed:(27)$$R\cdot Pi+T=P_{w}i\Rightarrow Pi=R^{-1}\cdot P_{w}i-T$$

Here, R is the rotation matrix of the camera relative to the world coordinate system and T is the translation vector of the camera in the world coordinate system. By solving Equation (27), the coordinates of the target feature points in the world coordinate system, $P_{i}$, can be obtained. At this point, the world coordinates of the target feature points, including depth values, are determined. By substituting this into Equations (20) and (21), a more accurate estimation of the projection error and radiation error in the VIO module, involving multiple vision and LiDAR sensors, can be obtained.

In each iteration, the system calculates the residual based on the current state estimate and observation data, expressed as *r* = *Hx* + *n*. In this equation, *H* is the measurement matrix, *x* is the state variable, and *n* is the noise. This residual is used to update the filter, refining the state variables and the covariance matrix. This process repeats until a stopping condition is met, such as reaching a predefined number of iterations, the residual falling below a certain threshold, or detecting a stable state.

The main advantages of the optimized VIO module are as follows:Based on the rapid response characteristics of the IMU, we employed IMU measurements to drive the process model, adapting to the maneuverability of composite robots.Leveraging the non-accumulative error attribute of stereo vision, we utilized localization estimates from stereo vision as observations in our observation model to correct errors in the IMU measurements.Considering the potential constraints of instantaneous sliding and jumping during the robot’s motion, we modeled the observed velocities on the *y*-axis and *z*-axis as zero-mean noise.

The entire workflow aims to utilize IMU data and stereo vision information. This allows for more accurate estimation and updating of the composite robot’s state. As a result, the system achieves more stable and reliable navigation and localization functionalities.

## 4. Experiments and Result

### 4.1. Experimental Platform

The experimental hardware platform discussed in this paper is a multifunctional composite robot, which includes an AGV (Automated Guided Vehicle) route chassis, a Duco brand collaborative robotic arm, and an integrated mapping and positioning system. The software is based on the ROS (Robot Operating System) platform, with chassis control managed by an embedded processor and control commands transmitted using the API (Application Programming Interface) protocol. The test hardware setup is illustrated in Figure 6. It features core components such as LIVOX brand LiDAR sensor, an RGB camera with an embedded IMU, and an infrared camera. The specific parameter information of the hardware sensors is shown in Table 1. All these components are integrated into the IIVL-LM system, which performs mapping and positioning in three-dimensional space. The platform’s mapping and localization performance was successfully tested in practical scenarios.

The experimental software testing environment was configured on a personal laptop equipped with the Ubuntu 20.04 operating system. This setup included integration with the ROS to facilitate the coordination of various module frameworks. To meet the high-performance requirements of the system applications being tested, the laptop was equipped with an Intel^®^ Core™ i5-10300H CPU running at a base clock speed of 2.50 GHz and an NVIDIA GeForce GTX 1650 Ti GPU (Graphics Processing Unit) with 8 GB of video RAM (Random Access Memory).

### 4.2. Experimental Environment and Dataset

The experiments were conducted in a simulated indoor rescue scenario covering an area of 484.3 square meters. The venue was not completely enclosed and included external sunlight penetration. To simulate varying illumination conditions, experiments were conducted at different time slots evenly distributed over a 24-h period. Additionally, local illumination changes were achieved through indoor lighting control. Luminance was measured using a lux meter, ranging from 10 to 50,000 lux. The composite robot accumulated a testing travel distance of 4792 m.

### 4.3. Experimental Evaluation Criteria

This paper assesses the X/Y axis errors and the variance between the theoretical positioning curve and the actual system positioning curve. Additionally, it primarily utilizes ATE to evaluate system accuracy. This method directly calculates the difference between the true values of the sensor poses and the estimated values from the positioning system. True values and estimates are aligned based on their timestamps, and then the difference between each pair of poses is calculated. This standard is particularly suited for evaluating the performance of positioning systems. The ATE for the *i*-th frame is defined as per Equation (28), where *Q* represents the true pose, *S* is the transformation matrix, and *P* is the estimated pose.
(28)$$F_{i}=Q_{i}^{-1}SP_{i}$$

The RMSE provides a more precise statistical measure of errors, as shown in Equation (29). In this equation, Δ represents the time interval, *n* denotes the total number of frames, m=n−Δ and $trans\left(E_{i}\right)$ corresponds to the translational component of the relative pose error.
(29)$$RMSE\left(F_{1:n},\Delta \right)={\left(\frac{1}{m}\sum_{i=1}^{m}{∥trans\left(F_{i}\right)∥}^{2}\right)}^{\frac{1}{2}}$$

### 4.4. Experimental Result

#### 4.4.1. Experiment on Illumination Changes in Simulated Indoor Rescue Scenarios

Due to the drift associated with odometry, we did not use it directly as an ‘absolute’ ground truth. Instead, we employed an independent third-party measurement method to determine the ground truth for validating the results, primarily using a laser tracker to ensure the reliability of the final evaluation. We first compared the ground truth curve with the self-estimated position curve obtained from the IIVL-LM system. One set of test data, conducted under standard indoor lighting conditions of 30,500 lux, is shown in Figure 7. After 10 tests, the average error on the X-axis was 0.42 cm, while on the Y-axis it was 0.46 cm. The mean squared error for the X-axis was 0.33%, and for the Y-axis, 0.54%.

Figure 8 illustrates the feature extraction results in the VIO module using RGB, infrared, and depth images under different lighting conditions in a small-scale indoor simulated environment. Key features of the environment can be effectively extracted from both the infrared images under low-light conditions and the RGB images in well-lit scenarios, including feature coordinates with depth values from the depth images. Through real-time illumination model conversion and nonlinear tight coupling, more precise loop detection and other related tasks can be achieved for keyframes.

Figure 9 presents the results of our real-time reconstruction of a small-scale indoor simulated environment, including detailed features of the environment, the robot’s pose, and the ground mobile robot’s path. Despite limited hardware resources, the mapping speed reached 3.79 ms per frame, demonstrating the method’s high real-time performance and efficiency under constrained conditions.

Furthermore, we utilized the same methodology to compare the IIVL-LM system with commonly used localization and mapping algorithms such as ORB-SLAM [23], VINS-Mono [45], R3LIVE++ [8], DSO [18], and SVO [27]. The comparisons are conducted under varying illumination conditions at different times. The specific results are presented in Table 2, and the operational routes are depicted in Figure 7c.

#### 4.4.2. Ablation Experiments on TUM-VI Dataset

The TUM-VI dataset comprises twenty-eight sequences captured in six different scenes, specifically including multiple sequences of photographs taken in indoor low-illumination and dark conditions. Without infrared image information, testing the multi-sensor fusion system’s reliability in dark situations using the algorithm proposed in this paper is challenging, as these sequences eventually lose visual features in RGB images. Table 3 presents the test results obtained from this dataset.

### 4.5. Experimental Analysis

This research primarily focuses on the fusion of infrared and RGB images for localization and mapping under changing lighting conditions or in darkness. Table 2 illustrates that, in contrast to mainstream methods, optimal outcomes were achieved under nighttime luminance levels between 20 and 8000 lux. Figure 10 shows that RMSE ATE values outperformed the second-best by 0.001 to 0.016. The accuracy of algorithms that do not incorporate LiDAR and instead rely only on sensors such as the IMU is severely affected in dark situations. Over a short distance of 910 m, the RMSE ATE values can rise to 0.044, which is a significant error resulting from the cumulative error and LiDAR offset caused by relying only on the IMU and LiDAR. Conversely, under well-lit conditions, the differences between the algorithms are not as pronounced. Experiments on the TUM-VI dataset proved that IIVL-LM did not gain an advantage when tested on a dataset without infrared image information, as shown in Figure 11. The results obtained by other methods were similar, proving the importance of an infrared camera as a visual input source in real-world dark environments.

### 4.6. Performance Testing of the Proposed Method Integrated into ORB-SLAM3

Although good conclusions were drawn from the analysis in the previous sections, it is important to note that R3LIVE++ and IIVL-LM use additional LiDAR input compared to other algorithms. To validate the effectiveness of our proposed method, we integrated the RGB and infrared fusion VIO module from this research into the open-source ORB-SLAM3 framework [24] for verification. Similarly, we tested different algorithms in a complex indoor environment with small-scale lighting variations. The test scenario is shown in Figure 12 and the results are shown in Table 4.

From Table 4, we can see that ORB-SLAM3 with our VIO module achieved good results in small-scale, complex indoor environments with varying levels of illumination. Particularly in low-light conditions, such as the case with a mean illumination of only 7110 lux at midnight, our proposed method demonstrated a significant advantage.

## 5. Conclusions and Discussion

We propose IIVL-LM, a multi-sensor, nonlinear, tightly-coupled framework for mapping and localization. The system integrates data from infrared cameras, IMUs, RGB cameras, and LiDAR to ensure that at least three sensors are operational under conditions of darkness or changing light. The system’s robustness is significantly increased by the integration, reducing drift issues and cumulative errors. Other sensory units frequently experience similar problems when insufficient data are available or when these conditions persist. We employ RGB image analysis to calculate real-time luminance, and by comparing this with maximum and minimum luminance levels, we generate a luminance conversion model. We adopt a dynamic nonlinear interpolation method based on luminance variations to couple sensors tightly at the data level and insert keyframes for loop closure detection. The system also performs a weighted fusion of infrared and RGB frames based on real-time luminance values, effectively ensuring the efficiency of feature extraction. Extensive comparative experiments were conducted with the composite robot in simulated scenarios such as unmanned dark factories and emergency rescue operations, achieving favorable results. However, there are still areas worthy of further research, detailed as follows:This paper primarily focuses on the selection and fusion of visual sensors based on luminance. These sensors are then input into the R3LIVE++ fusion framework as part of the Vision SLAM module. This aspect requires further in-depth research and optimization.This method has been tested and validated only on ground-based composite robots. However, rescue operations are often three-dimensional, and the system’s infrared camera supports 3D depth detection. Future applications in three-dimensional rescue, automated factory settings, and other areas will inevitably require the integration of unmanned aerial vehicles with ground robots. This integration is necessary to achieve 3D mapping, localization, and navigation.Continued in-depth research in environmental modeling and cognition is necessary. Traditional environmental modeling methods are insufficient for robots’ autonomous navigation in unknown and complex environments, necessitating advanced approaches for a deeper understanding of the environment. Future studies should consider a top-down, model-based approach, employing techniques such as Conditional Random Fields and Markov Random Fields to articulate scene positional data, scale information, inter-object relationships, and the probabilities of specific objects’ presence within the scene. This strategy enables rapid scene evaluation based on limited information, mimicking human-like perception abilities.Research on the autonomous learning of behaviors should also be expanded. Using machine learning techniques like reinforcement learning and integrating terrain understanding at the feature level is advantageous when designing behavior controllers. This online learning enhances the behavior controllers’ adaptability and flexibility in unfamiliar environments.

## Figures and Tables

**Figure 1 sensors-24-07381-f001:**
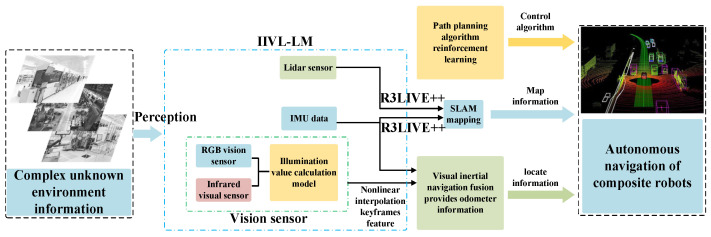
IIVL-LM system framework applied to the composite robot.

**Figure 2 sensors-24-07381-f002:**
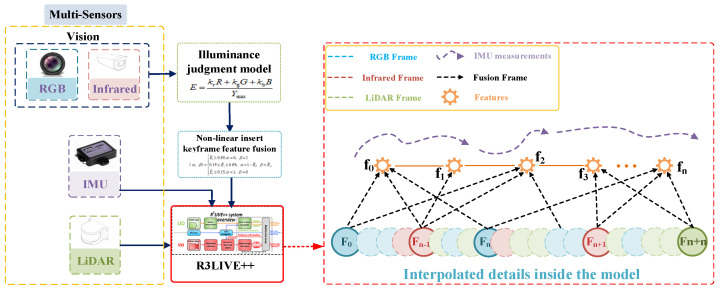
Schematic diagram of each module of the IIVL-LM system.

**Figure 3 sensors-24-07381-f003:**
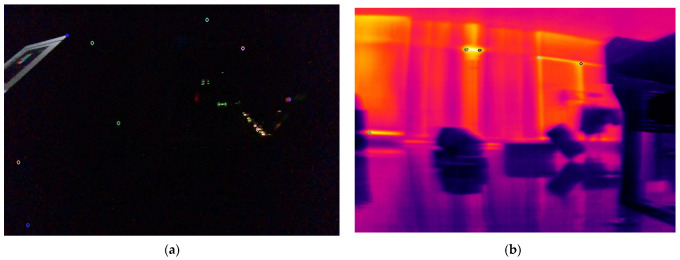
Feature extraction performance of RGB and infrared images under extreme illuminance value. (**a**) Feature extraction performance of RGB images at a normalized illuminance value of 0.148. (**b**) Feature extraction performance of infrared images at a normalized illuminance value of 0.853.

**Figure 4 sensors-24-07381-f004:**
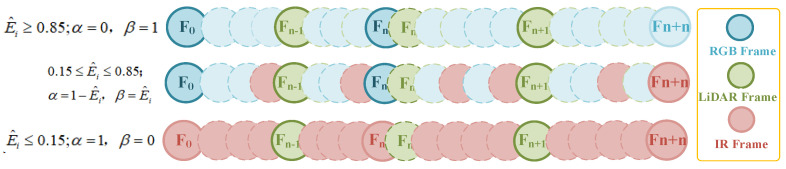
Weight-based nonlinear interpolation frame method.

**Figure 5 sensors-24-07381-f005:**
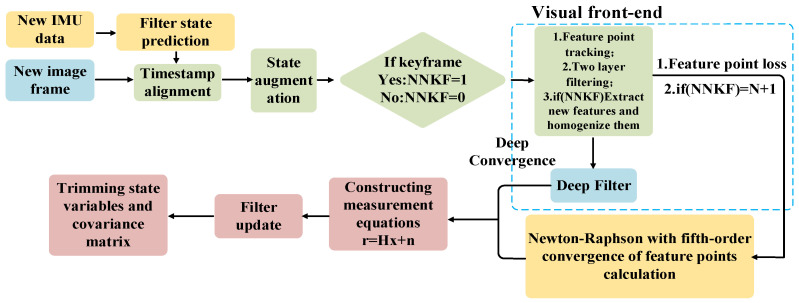
VIO (Optimized Visual-Inertial Odometry).

**Figure 6 sensors-24-07381-f006:**
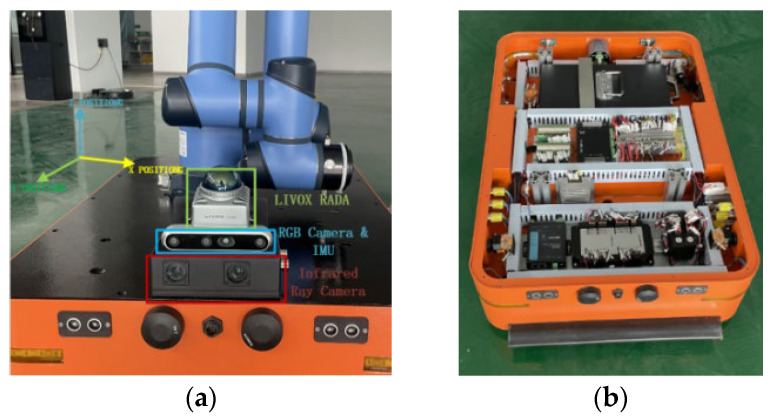
Schematic diagram of IIVL-LM system and sensors deployed on composite robots. (**a**) Multi-sensor. (**b**) Composite robots.

**Figure 7 sensors-24-07381-f007:**
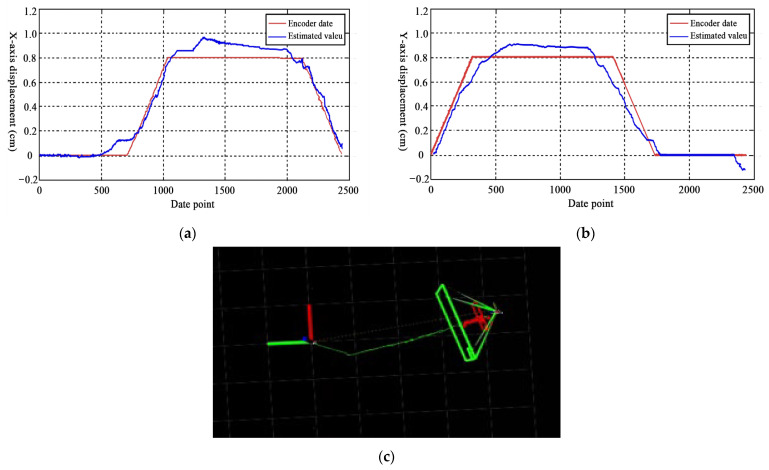
Comparison of X/Y axis data and actual trajectory of review robots under the IIVL-LM system. (**a**) Comparison of *X*-axis data. (**b**) Comparison of *Y*-axis data. (**c**) Actual testing and running trajectory of composite robots.

**Figure 8 sensors-24-07381-f008:**
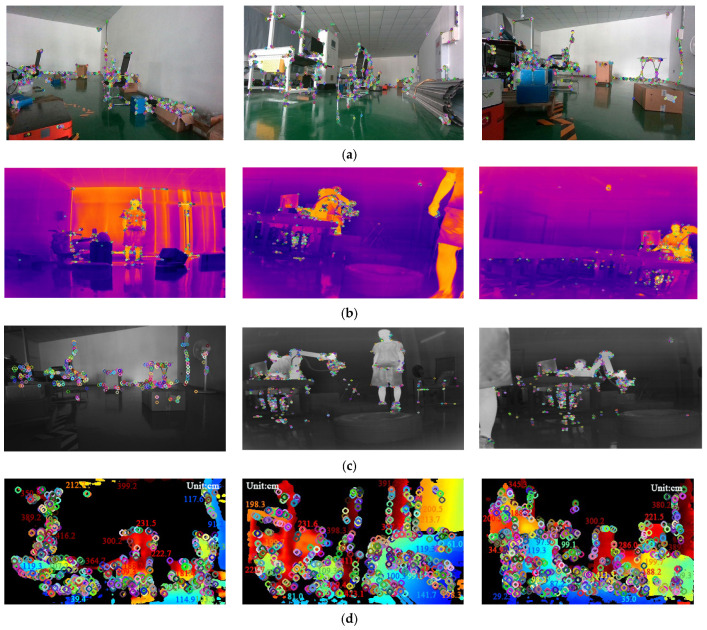
The feature extraction results in the VIO module using RGB, infrared, and depth images under different lighting conditions in a small-scale indoor simulated environment. (**a**) The extraction of environmental features from RGB frames during the day. (**b**) Feature extraction of environmental characteristics from infrared frames during the day. (**c**) Feature extraction of environmental characteristics from infrared frames during the night. (**d**) Feature coordinates with depth values in the depth image.

**Figure 9 sensors-24-07381-f009:**
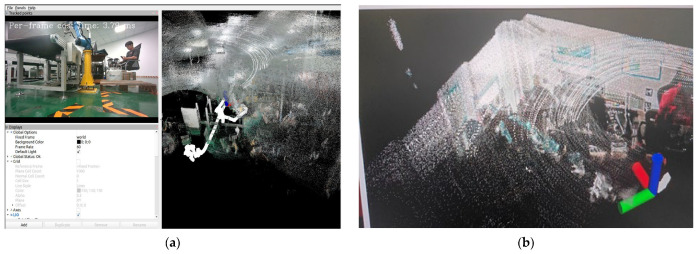
Real-time reconstruction process and radiance map of the small-scale indoor environment. (**a**) Real-time reconstruction process of the map. (**b**) Reconstructed radiance map of the small-scale indoor environment.

**Figure 10 sensors-24-07381-f010:**
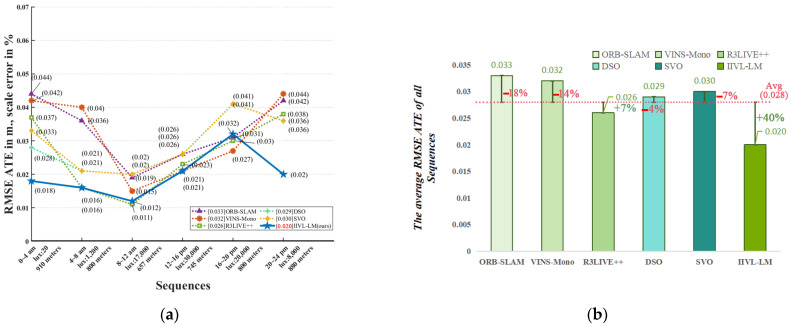
Test conclusion and comparison under different illuminances. (**a**) RMSE ATE of all methods under different illuminance values. (**b**) Comparison between various methods and overall average.

**Figure 11 sensors-24-07381-f011:**
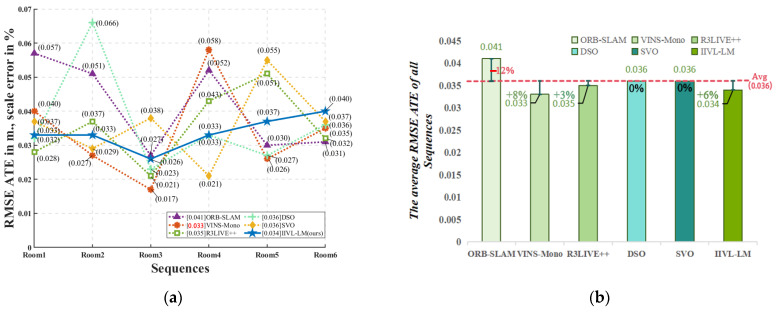
Test conclusion and comparison under multiple sequences in the TUM-VI dataset. (**a**) RMSE ATE of all methods under multiple sequences in the TUM-VI dataset. (**b**) Comparison between various methods and overall average.

**Figure 12 sensors-24-07381-f012:**
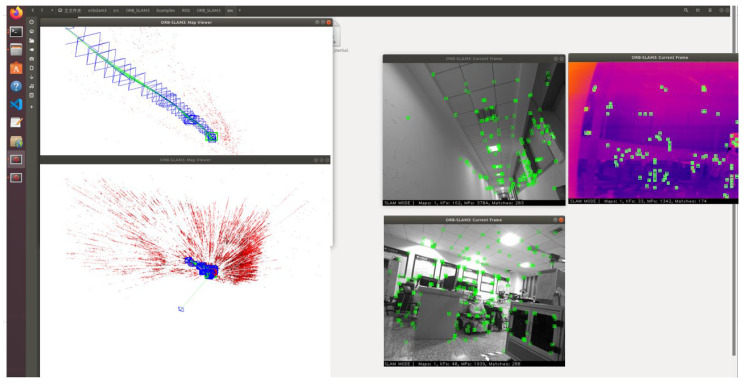
The test scenario on ORB-SLAM3.

**Table 1 sensors-24-07381-t001:** The main parameters of the selected sensors for the experiment.

No.	Sensors	Items	Parameter Value
1	LiDAR	Wavelength	905 nm
		FOV	Horizontal 360°, vertical −7~52°
		Electrical cloud output	200,000 Points/second, 10 Hz
		Point cloud frame rate	10 Hz
		Near blind spots	0.1 m
		Measurement distance and accuracy	0.06 to 10 m, Max.30 m, 270° ±40 mm
		Data synchronization method	IEEE 1588-2008 (PTPv2), GPS
2	RGB Camera	Resolution	752 × 480
		Perspective and applicable distance	D:140° H:120° V:75°, 0.8–10 m
		Maximum frame rate	60 FPS
3	IMU	frequency	100~500 Hz
4	Infrared Camera	Effective IR distance	3 m
		Frame rate	60 FPS

**Table 2 sensors-24-07381-t002:** Comparison of performance of various methods under different illumination levels (RMSE ATE in m., scale error in %).

Time Frame	The Average Illumination Value (lux)	Distance (m)	ORB-SLAM	VINS-Mono	R3LIVE++	DSO	SVO	IIVL-LM (Ours)
0~4 a.m.	22	910	0.044	0.042	0.037	0.028②	0.033	0.018①
4~8 a.m.	1211	800	0.036	0.040	0.016①	0.021	0.018②	0.016①
8~12 a.m.	17,490	657	0.019	0.015	0.011①	0.020	0.018	0.012②
12~16 p.m.	29,782	745	0.026	0.021①	0.023②	0.026	0.032	0.021①
16~20 p.m.	20,034	800	0.031	0.027①	0.030②	0.041	0.039	0.032
20~24 p.m.	7980	880	0.042	0.044	0.038	0.036②	0.037	0.020①
Avg RMSE ATE			0.033	0.032	0.026②	0.029	0.030	0.020①
Compare with the population average (0.028)			−18%	−14%	+7%②	−4%	−7%	+40%①

①② represent the top 2 results for the algorithm.

**Table 3 sensors-24-07381-t003:** Comparison of performance of various methods with TUM-VI dataset (RMSE ATE in m., scale error in %).

Seq.	ORB-SLAM	VINS-Mono	R3LIVE++	DSO	SVO	IIVL-LM (Ours)
Room 1	0.057	0.040	0.028①	0.032②	0.037	0.033
Room 2	0.051	0.027①	0.037	0.066	0.029②	0.033
Room 3	0.027	0.017①	0.021②	0.023	0.038	0.026
Room 4	0.052	0.058	0.043	0.033②	0.021①	0.033②
Room 5	0.030	0.026①	0.051	0.027②	0.055	0.037
Room 6	0.031①	0.035	0.032②	0.036	0.037	0.040
Avg RMSE ATE	0.041	0.033①	0.035	0.036	0.036	0.034②
Compare with the population average (0.036)	−12%	+8%①	+3%	0%	0%	+6%②

①② represent the top 2 results for the algorithm.

**Table 4 sensors-24-07381-t004:** Comparison of the performance of the proposed VIO module integrated into ORB-SLAM3 with other visual SLAM algorithms under different illumination conditions (RMSE ATE in m., scale error in %).

Time Frame	The Average Illumination Value (lux)	Distance (m)	ORB-SLAM	VINS-Mono	DSO	SVO	ORB-SLAM3	ORB-SLAM3 (with Ours VIO Module)
0~4 a.m.	184	460	0.021	0.028	0.030	0.021	0.017②	0.012①
4~8 a.m.	1532	600	0.029	0.023②	0.037	0.033	0.024	0.017①
8~12 a.m.	19,330	400	0.013	0.008①	0.011	0.012	0.009②	0.010
12~16 p.m.	30,414	500	0.019	0.020	0.018	0.014①	0.016②	0.017
16~20 p.m.	22,615	480	0.014	0.010①	0.018	0.011②	0.012	0.012
20~24 p.m.	7110	510	0.028	0.025	0.022②	0.030	0.039	0.017①
Avg RMSE ATE			0.021	0.019②	0.023	0.020	0.020	0.014①
Compare with the population average (0.020)			−5%	+5%②	−15%	0%	0%	+30%①

①② represent the top 2 results for the algorithm.

## Data Availability

The data analyzed or generated in this study are available from the corresponding authors upon reasonable request.

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
