# Peer review of "A Novel Multi-Sensor Nonlinear Tightly-Coupled Framework for Composite Robot Localization and Mapping"

_sensors, 2024, doi:10.3390/s24227381_

Round 1
Reviewer 1 Report
Comments and Suggestions for Authors
Please see the attached file.

Author Response
Please see in the attachment.

Reviewer 2 Report
Comments and Suggestions for Authors
The manuscript proposed infrared measurements with a luminance model and a weighted feature fusion method. The authors claimed that the framework improved RMSE error in a simulated indoor rescue environment.
The following are my concerns:
1. In Eq 16, how were the numbers 0.15 and 0.85 determined to make sure the weighted fusion in this way would work in general environments?
2. Considering the drift involved in odometry, it should not be used as ground truth.
3. In Table 2, were all the methods listed tested with IMU and infrared measurements? The proposed contribution is the method for integrating infrared measurements. From the table, R3LIVE++ is already better than most other methods in the setting, and the IIVL-LM mainly gains performance improvement when lux is low. The results are naturally expected since more information is considered.
To verify the effectiveness of the proposed method, the authors should also integrate it into other open-sourced solutions or improve its generality theoretically.
Author Response
Please see in the attachment.
